# Patterns of adherence to home blood pressure monitoring among men and women in the Electronic Framingham Heart Study

Tenes J. Paul[1], Katherine Sadaniantz[1], Apurv Soni[1], Jean-Claude Asaker[1], Chathurangi H. Pathiravasan[2], Jordy Mehawej[1], Andreas Filippaios[1], Yuankai Zhang[3], Ziyue Wang[1], Chunyu Liu[3], Honghuang Lin[1], Joanne M. Murabito[4,5], David D. McManus[1], Lara C. Kovell[1]*

1 Department of Medicine, UMass Chan Medical School, Worcester, Massachusetts, United States of America, 2 Department of Biostatistics, Johns Hopkins Bloomberg School of Public Health, Baltimore, Maryland, United States of America, 3 Department of Biostatistics, Boston University School of Public Health, Boston, Massachusetts, United States of America, 4 Boston University and The NHLBI's Framingham Heart Study, Framingham, Massachusetts, United States of America, 5 Department of Medicine, Division of General Internal Medicine, Boston Medical Center, Boston University Chobanian and Avedisian School of Medicine, Boston, Massachusetts, United States of America

* lara.kovell@umassmemorial.org

## Abstract

Worldwide, there are differences in hypertension control by sex. The use of home blood pressure monitoring (HBPM) is associated with lower average blood pressures and higher medication adherence. However, little is known about adherence trajectories and sex differences in HBPM. This study characterizes adherence to HBPM among those with and without hypertension. Electronic Framingham Heart Study participants were instructed to perform HBPM weekly for 1 year. Adherence was defined as having ≥1 measurement per week averaged over 4-week segments. Primary exposures of hypertension status and sex were self-reported. Group-based trajectory modeling was used to identify adherence trajectories. Logistic regression was applied to investigate factors associated with membership in each trajectory group in the sex-stratified and whole cohorts. Among 990 participants (58% women, age 53 ± 9 years, 26% hypertension), three groups with distinct HBPM adherence patterns were identified: "early discontinuation", "gradual decrease", and "high adherence". Participants with hypertension were more likely to maintain "high adherence" compared to those without (OR 1.55; 95% CI 1.08–2.23), with similar findings seen among women with hypertension (OR 2.24; 95% CI 1.35–3.72) but not men. In women, these findings remained significant when adjusting for anxiety, depression, and blood pressure, but were attenuated by adjusting for age and income. This study highlights HBPM adherence trajectories and shows that women with hypertension were more likely to be in the high adherence group, though these associations were attenuated after

**Data availability statement:** Data cannot be shared publicly because access to these data is managed the Framingham Heart Study Repository. Data are available from the Framingham Heart Study for researchers who meet the criteria for access to confidential data. They can be contacted through a web-based application at https://www.framinghamheartstudy.org/fhs-for-researchers/research-application/.

**Funding:** "The Framingham Heart Study (FHS) was financially supported by the National Heart, Lung, and Blood Institute (NHLBI) of the National Institutes of Health and Boston University School of Medicine in the form of an NIH award (75N92019D00031). The electronic FHS research study was financially supported by a grant (R01HL141434) and an award (Robert Wood Johnson Award). This study was also financially supported by the National Heart, Lung, and Blood Institute (NHLBI) in the form of a grant (K23HL163450) received by LK. This study was also financially supported by the National Heart, Lung, and Blood Institute (NHLBI) in the form of grants (R01HL155343, R01HL141434, R33HL158541, U54HL143541 and U54HL143541-05S1, and UG3NS135168) received by DDM. This study was also financially supported by the National Heart, Lung, and Blood Institute (NHLBI) in the form of grants (U54HL143541 and U54HL143541-05S1, U01HL146382, and UG3NS135168) received by AS. This study was also financially supported by the American Heart Association in the form of a grant (AHA_18SFRN34110082) received by AF.".

**Competing interests:** "The authors have read the journal's policy and have the following competing interests: DDM has received research support from Fitbit, Apple Inc, Bristol–Myers Squibb, Boehringer–Ingelheim, Pfizer, Flexcon, Samsung, Philips Healthcare, and Biotronik, and he has received consultancy fees from Heart Rhythm Society, Bristol–Myers Squibb, Pfizer, Fitbit, Flexcon, Boston Biomedical Associates, VentureWell, Avania, NAMSA and Rose Consulting. DDM also declares financial support for serving on the Steering Committee for the GUARD-AF study (NCT04126486) and the advisory committee for the Fitbit Heart study (NCT04176926). AS has received consultancy fees from Pfizer and Advanced Research Program Agency for Health (ARPA-H). This does not alter our adherence to PLOS ONE policies on sharing data and materials. There are no patents, products in development or marketed products associated with this research to declare".

adjusting for demographic factors and co-morbidities. Future studies should explore strategies to enhance adherence in populations at risk of early discontinuation.

## Introduction

Hypertension is one of the most common chronic diseases worldwide with an estimated prevalence of 1.3 billion [1]. In the United States alone, 122.4 million adults (47% of the population) meet the criteria for a diagnosis of hypertension [2], with 36% of the population recommended to be on antihypertensive medications [3,4]. Despite hypertension being a major risk factor for ischemic heart disease—the leading cause of death worldwide—it is estimated that 46% of individuals with hypertension remain unaware and untreated [5]. While tight blood pressure control has been shown to decrease the risk of cardiovascular events and all-cause mortality, rates of control in adults with hypertension have been inadequate [6,7]. National Health and Nutrition Examination Survey (NHANES) data found that despite hypertension control improving in the first ten years of the 21st century, the subsequent ten years have seen a more modest increase, with some groups, including women, experiencing declines [2]. Similar findings were seen in women with hypertension in the Canadian Health Measures Survey, with declining treatment and control rates in recent years [8]. With advancing age, the prevalence of hypertension among women surpasses that of men, with large cohorts reporting lower control rates and widening disparities with age in women vs. men despite higher rates of hypertension awareness and seeking treatment [8,9]. Moreover, women have higher rates of white coat hypertension, further complicating diagnosis and management [9–11].

Novel technologies, such as mobile health interventions, are associated with significant reductions in blood pressure, thereby offering a potential solution to addressing this sex-related disparity in hypertension management [12]. Home blood pressure monitoring (HBPM) in particular is indicated to confirm the initial diagnosis of hypertension, response to treatment, and for evaluation of white coat hypertension [4,13,14]. The use of HBPM, even at a frequency of weekly or monthly, has been associated with lower average blood pressure and higher adherence to antihypertensive medications [15,16], with effects that span across different cultures and demographic groups [17]. Despite this, adherence to HBPM is challenging due to a variety of factors, including patient motivation, health literacy issues, or inadequate training and guidance from healthcare providers [18–20]. Given the known sex differences in blood pressure trajectories, control, and white coat hypertension, there are many factors that can contribute to sex differences in HBPM [21–24]. A study of NHANES data from 2011–2014 showed no difference in HBPM use by sex, though HBPM frequency increased with advancing age, healthcare utilization, and having health insurance.[10] However, other studies have shown sex-differences in HBPM, though through self-reporting rather than device-based reporting of adherence [24–27]. The ambiguity surrounding sex-based differences in adherence to HBPM presents the opportunity to gain insight into the use of this tool [16,25,26,28–30].

In this study, we analyzed participants from the electronic Framingham Heart Study (eFHS) who were asked to regularly monitor their blood pressure at home. We aimed to characterize adherence to HBPM by sex in those with and without hypertension, while determining other factors associated with greater HBPM adherence. Based on the prior literature, we hypothesize that women and older adults are more likely to adhere to HBPM.

## Materials and methods

### Data source: Electronic Framingham Heart Study cohort

The present study is a prospective cohort study analyzing data collected from participants enrolled in eFHS. The development of the eFHS cohort has been previously reported [31,32]. In brief, eFHS is an electronic sub-cohort of the Framingham Heart Study composed of members of the Third Generation Cohort, the New Offspring Cohort, and the multiethnic Omni Group 2 Cohort. These cohorts are the third generation of the original Framingham Heart Study cohort, the offspring of the original Framingham Heart Study cohort and their spouses, and members of the Framingham community from underrepresented ethnic groups, respectively [31,32]. The eFHS protocol was approved by the Institutional Review Board of Boston University Medical Center. This study has been approved by the appropriate ethics committee and has therefore been performed in accordance with the ethical standards laid down in the 1964 Declaration of Helsinki and its later amendments. All participants gave their informed consent prior to their inclusion in the study.

The eFHS cohort enrolled participants from the aforementioned cohorts starting between 2016 and 2019 by inviting them during regularly scheduled Framingham Heart Study research center examinations. To be eligible, participants had to own a smartphone (an iPhone with iOS version 9 or higher), be able to communicate in English, reside in the United States, and be willing to provide permissions for data sharing and notifications with research study staff. Android users were excluded from this study as the digital blood pressure device did not pair with the Android phone. Participants were given an Apple Watch series 0 as well as a Nokia-Withings Digital blood pressure device model BP-801 [31]. This blood pressure device was chosen given its Food and Drug Administration approval for HBPM, accuracy ±3 mmHg, deployment in other digital health studies, and validation via the international scientific organization, Stride BP, which provides lists of validated blood pressure measuring devices [33,34]. All participants were instructed on the use of both electronic devices, including an initial in-person demonstration of the appropriate use of the blood pressure device, when possible. They were given written instructions on the use of their blood pressure device and on techniques for proper HBPM [31].

### Study protocol

Using the provided digital blood pressure devices, participants were instructed to check their blood pressure once weekly on the same day and around the same time over the course of the one-year follow-up period. All participants with at least one recorded blood pressure reading were included in this study. Blood pressure readings were synchronized with the eFHS app through an associated smartphone app (Nokia Health Mate). Blood pressure data were securely transmitted to the research center via the eFHS app. Any additional blood pressure readings collected at the participants' discretion were also collected and sent to the research center [31]. The initial blood pressure reading collected in the research center on the day of enrollment to teach participants how to properly use the system was excluded.

### Group-based trajectory modeling

Group-based trajectory modeling was used to identify trajectory patterns of adherence among the sample and was run using SAS with 2–5 groups. Bayesian Information Criterion (BIC) scores were calculated for different numbers of groups. The SAS procedure "Proc TRAJ" was used for group-based trajectory modeling. We assessed model fit by comparing the BIC of the 2-group and 3-group models. The 2-group model had a group that maintained strong adherence throughout the study period and a second group that had poor adherence up front and never improved. The 3-group model had an

intermediate group who gradually decreased their adherence over the year. Despite not having the best BIC score, the final model with three distinct trajectories was selected as it accounted for a third, intermediate group – a closer reflection of the typical trajectories real-life patients may have. This model is more inclusive of the types of adherence frequently encountered in clinical practice. In a sensitivity analysis, a 2-group model (high vs. low adherence) produced an odds ratio in the same direction as in the 3-group model, but did not reach statistical significance (S1 Table). This further supports the use of a 3-group classification, which provides more granular analysis to detect meaningful patterns.

### Definitions of exposure and outcome

To assess the role of sex-based differences between each of trajectory groups, a primary exposure of sex (per the eFHS survey, either male or female) was designated. A diagnosis of hypertension, based on participants' response ("yes"/"no") from a survey question in the eFHS examination data: "have you been told by your doctor you have high blood pressure or hypertension," was selected as an effect modifier. These data were collected at the time of enrollment during an in-person examination. HBPM adherence was measured over the 52-week follow-up period, broken up into 4-week segments. The primary outcome was adherence to weekly HBPM. For each week, participants were coded as "0" or "1" – with "0" indicating that they did not log a HBPM reading and "1" if they had logged at least one reading. HBPM adherence was defined as ≥1 blood pressure measurement per week, averaged over 4-week segments. For example, if participants logged one or more blood pressure readings for three weeks but did not log a reading in the fourth week, their weekly HBPM adherence was calculated as 75%.

### Covariates

The covariates in the analysis were selected based on a qualitative assessment of what factors might be most clinically relevant and were collected in-person at the examination center unless specified. Among those, covariates found to be statistically significant in the univariate analysis ($p < 0.05$) and those that had a potentially clinically relevant association with HBPM use were selected in the models. Age as a continuous variable was included, given that hypertension prevalence increases with advancing age and older individuals tend to demonstrate higher rates of adherence [30]. Similarly, household income, defined as a categorical variable (<$35,000/year, $35,000–100,000/year, and>$100,000/year) was included given that individuals with higher socioeconomic status may have higher rates of self-advocacy and may be more medically literate [35]. Baseline systolic blood pressure from the initial home blood pressure measurement was included given that having a higher blood pressure can impact motivation for adherence to HBPM. Finally, baseline anxiety and depression were included, as these conditions—particularly when coinciding with one's medical conditions—may influence frequency of HBPM [36]. Mental health conditions were self-reported: participants were asked "Have you ever been told that you have any of the following mental health conditions?" and were given the choices: "depression," "anxiety," other mental health diseases," or "none of the above." Models were progressively adjusted as follows: Model 1 adjusted for age, Model 2 adjusted for age and income, and Model 3 adjusted for age, baseline systolic blood pressure, and anxiety. Models 1 and 2 were aimed at evaluating demographic covariates while Model 3 evaluated clinical factors that may influence adherence.

### Statistical analysis

Statistical significance was considered a two-sided P value <0.05 for all statistical analyses. All analyses were performed using SAS version 9.4. Descriptive statistics were calculated for continuous variables using means and standard deviations and for categorial variables using frequency counts and percentages. The difference between adherence groups was assessed using the Chi-square test and ANOVA test for categorical variables and continuous variables, respectively. Given our focus on adherence trajectory differences by sex, we first used logistic regression models in the overall study

cohort to evaluate the outcome HBPM adherence. In these models, we included an interaction term between hypertension status (yes vs. no) and sex to examine whether adherence patterns differed by both hypertension status and sex. We then stratified by sex, based on an *a priori* decision, and used multivariable logistic models to investigate the association of the adherence patterns with hypertension status. These stratified models were adjusted for age, income, anxiety, and systolic blood pressure using the three models described above. Any observations that were missing a value for any variable were discarded from the analysis.

## Results

### Study cohort

Among the 1918 eFHS enrollees, 1125 participants chose to use a digital blood pressure device. Those who did not record at least 1 blood pressure reading were excluded, leaving 990 participants. Of the total study cohort (N = 990), 58.1% were women and the mean age (±SD) was 53 ± 9 years. The median follow-up was 375 days from the enrollment date. At baseline, 25% of participants reported a diagnosis of hypertension and 21.3% (N = 211) were being treated with antihypertensive medications as shown in Table 1. Most participants (93%) self-identified as non-Hispanic White. The majority (76%) were married and had completed at least some college (91%), with 61% reporting annual household income > $100,000. Among the study cohort, there were relatively low prevalences of diabetes and cardiovascular disease at 8% and 4%, respectively (Table 1).

### Trajectories of adherence over the study period

In developing the adherence trajectory groups, a two-group model had the lowest BIC score (BIC = −9485.4 vs −8633.5 for the three-group model we ultimately used), however, the two-group model showed one group that maintained very high adherence throughout the study period and one that immediately discontinued HBPM.

The three trajectory groups correspond to distinct patterns of HBPM adherence (Fig 1) and were named based on their adherence characteristics. The largest group, "early discontinuation" (N = 429, 43.3% of total, 23.4% hypertension), demonstrated an early decline in adherence from the beginning of the study period. Adherence to weekly monitoring began at 48.0% and had a steep initial drop-off, decreasing to 1.7% by 28 weeks (study halfway point). The second trajectory group, "gradual decrease" (N = 339, 34.3% of total, 24.5% hypertension), began with 46.2% adherence to weekly monitoring and completed the 52 weeks at 28.5% adherence. The final trajectory group, "high adherence," (N = 222, 22.4% of total, 32.1% hypertension) began with 79.9% adherence to weekly monitoring, which decreased to 75.9% at the end of 52 weeks.

### Trajectory group characteristics

When examining the three trajectory groups in univariate analysis, participants in the high adherence group were on average older than those in the early discontinuation and gradual decrease groups, (mean ages 56.6, 51.4, and 53.2 years, respectively; *p* < 0.001) shown in Table 2. There were no significant differences between the groups related to education level or full-time employment. In an unadjusted analysis of the entire study cohort, hypertension was associated with higher odds of high adherence versus early discontinuation (unadjusted OR 1.55; 95% CI 1.08–2.23) (S2 Table). However, this association was attenuated and no longer significant after adjustments for age, income, anxiety/depression, and baseline systolic blood pressure.

### Sex-based differences across the trajectory groups

In the overall population logistic regression models evaluating factors associated with HBPM adherence, we examined the interaction term between hypertension status and sex and found it to be non-significant (*p* = 0.11). In the sex-stratified

**Table 1. Baseline characteristics of women and men, stratified by diagnosis of hypertension.**

| | Total Sample (n = 990)* | Women (n = 576) | | Men (n = 414) | |
|---|---|---|---|---|---|
| | | Hyperten-sion (n = 126) | No hyperten-sion (n = 448) | Hyperten-sion (n = 128) | No hyperten-sion (n = 286) |
| *Sociodemographic and biometric variables* | | | | | |
| Age, years | 53.2 ± 8.6 | 56.4 ± 7.7 | 51.7 ± 8.4 | 57.6 ± 8.1 | 52.1 ± 8.4 |
| **Race and Ethnicity** | | | | | |
| Non-Hispanic Black | 21 (2.1) | 3 (2.4) | 9 (2.0) | 5 (3.9) | 4 (1.4) |
| Non-Hispanic White | 913 (93.0) | 117 (92.9) | 420 (94.8) | 115 (90.6) | 259 (91.2) |
| Hispanic | 25 (2.6) | 5 (4.0) | 6 (1.4) | 6 (4.7) | 8 (2.8) |
| Married/living as married | 747 (76.2) | 85 (68.6) | 331 (74.2) | 106 (83.5) | 223 (79.1) |
| ≥ Some college education | 901 (91.4) | 105 (83.3) | 417 (93.5) | 107 (84.3) | 270 (94.7) |
| Full-time employment | 700 (71.1) | 76 (60.3) | 290 (64.9) | 86 (68.3) | 246 (86.6) |
| **Annual household income** | | | | | |
| Low (<$35,000) | 50 (5.4) | 10 (8.7) | 24 (5.7) | 4 (3.4) | 11 (4.0) |
| Mid-range ($35,000-$100,000) | 308 (33.2) | 44 (38.3) | 152 (36.4) | 47 (39.8) | 65 (23.7) |
| High (>$100,000) | 569 (61.4) | 61 (53.0) | 242 (57.9) | 67 (56.8) | 198 (72.3) |
| *Clinical and biometric factors* | | | | | |
| Current smoking | 67 (6.8) | 9 (7.1) | 32 (7.1) | 6 (4.7) | 20 (7.0) |
| Anti-hypertensive medication use | 211 (21.3) | 101 (80.2) | 3 (0.7) | 106 (82.8) | 1 (0.4) |
| Diabetes diagnosis | 77 (7.8) | 21 (16.7) | 14 (3.1) | 27 (21.1) | 15 (5.2) |
| Cardiovascular disease | 40 (4.0) | 8 (6.4) | 4 (0.9) | 14 (10.9) | 13 (4.6) |
| Atrial fibrillation | 18 (1.8) | 2 (1.6) | 4 (0.9) | 4 (3.1) | 8 (2.8) |
| Renal disease | 22 (2.2) | 4 (3.2) | 7 (1.6) | 7 (5.5) | 4 (1.4) |
| Body mass index (kg/m$^2$) | 27.6 ± 4.9 | 29.1 ± 5.3 | 26.0 ± 4.7 | 31.0 ± 4.7 | 28.1 ± 3.9 |
| *Self-reported activity, health, and mental health* | | | | | |
| Sedentary hours | 8.2 ± 3.1 | 7.7 ± 3.5 | 8.0 ± 3.1 | 8.6 ± 3.2 | 8.6 ± 3.1 |
| Moderate/heavy physical activity, hours | 2.9 ± 2.2 | 2.9 ± 2.3 | 2.7 ± 2.0 | 3.4 ± 2.5 | 3.1 ± 2.4 |
| Self-reported excellent health | 262 (26.5) | 18 (14.3) | 144 (32.1) | 18 (14.1) | 82 (28.7) |
| Depression | 155 (16.5) | 34 (27.4) | 75 (17.5) | 15 (12.6) | 31 (11.4) |
| Anxiety | 134 (14.2) | 25 (20.2) | 80 (18.7) | 9 (7.6) | 20 (7.4) |
| *Blood pressure characteristics* | | | | | |
| Baseline Systolic BP | 120 ± 15 | 126 ± 15 | 114 ± 13 | 131 ± 14 | 121 ± 12 |
| Baseline Diastolic BP | 76 ± 9 | 77 ± 9 | 73 ± 8 | 81 ± 9 | 79 ± 8 |
| Mean daytime home SBP† | 123 ± 13 | 126 ± 12 | 116 ± 11 | 133 ± 11 | 126 ± 11 |
| Systolic BP variability‡ | 0.10 | 0.09 | 0.10 | 0.09 | 0.09 |
| Diastolic BP variability‡ | 0.11 | 0.10 | 0.10 | 0.13 | 0.10 |
| Number of weeks with ≥ 1 home BP measurements | 17.7 (15.3) | 20.3 (15.9) | 16.3 (14.3) | 18.4 (16.1) | 18.2 (15.9) |

Abbreviations: BP – blood pressure.

* Data presented as mean±SD or N (%).

† Daytime defined as 5am-5 pm, nighttime defined as 5 pm-5am.

‡ Coefficient of variability (standard deviation/mean).

unadjusted model, when compared to women without hypertension, women with hypertension were 2.24 times more likely to be in the high adherence group vs. early discontinuation (unadjusted OR 2.24; 95% CI 1.35–3.72) as seen in Table 3. This association was attenuated by adjustments for age and income in Models 1 and 2 while the association persisted in

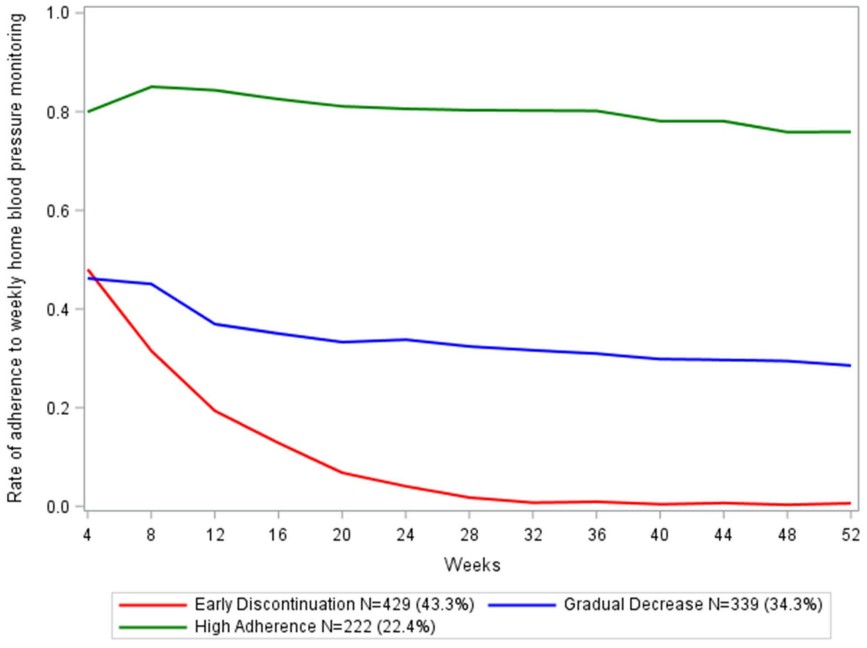

**Fig 1. Three distinct trajectory groups illustrating adherence to HBPM over 1 year.**

models adjusting for age, anxiety/depression, and baseline blood pressure. No associations were observed in the gradual discontinuation vs. early discontinuation for all the models.

When compared to men without hypertension, men with hypertension did not have higher odds of being in the high adherence group vs. early discontinuation in either the unadjusted or adjusted analysis (unadjusted OR 1.05; 95% CI 0.62–1.76). Further comparison between men and women with and without hypertension is provided in S3 Table.

## Discussion

In this middle-aged and older adult cohort of Framingham Heart Study participants who monitored home blood pressure as part of a study protocol, three distinct trajectory groups of early discontinuation, gradual decline, and high adherence were observed. The odds of achieving a high adherence trajectory, as opposed to early discontinuation, was significantly higher among the entire cohort and women with hypertension compared to those without hypertension. Among women, these associations were attenuated when adjusting for age and income, and remained significant when adjusting for anxiety/depression and baseline systolic blood pressure. Notably, the same association was not seen among men with hypertension. Although the interaction term between hypertension status and sex was non-significant, the differences in adherence patterns by sex and hypertension status in the stratified analyses are worth further exploration in larger studies and studies investigating alternative HBPM schedules.

These findings may expose gaps in the utilization of HBPM among men, leading to suboptimal adherence over time. Prior studies have examined sex-related factors associated with home blood pressure monitoring. The ESTEBAN study, a cross-sectional study of French adults, found that being a woman was associated with having a blood pressure monitoring device at home [37]. A study from Poland found that women, compared to men, were significantly more likely to be given a blood pressure monitor, perform HBPM regularly, and keep a blood pressure diary [25]. Our findings build on these prior studies and raise the question of why men may be less likely to maintain high adherence to HBPM. These data present

**Table 2. Univariate characteristics of the three home blood pressure monitoring adherence trajectory groups identified by group-based trajectory modeling.**

| | Early discontinuation (n = 429)* | Gradual decrease (n = 339)* | High adherence (n = 222)* | *p*-value |
|---|---|---|---|---|
| Women, total | 246 (57.3) | 209 (61.7) | 121 (54.5) | 0.22 |
| All participants with Hypertension | 100 (23.4) | 83 (24.5) | 71 (32.1) | **0.04** |
| Women with hypertension | 42 (9.8) | 46 (13.6) | 38 (17.2) | **0.04** |
| Men with hypertension | 58 (13.6) | 37 (10.9) | 33 (14.9) | |
| Women without hypertension | 203 (47.4) | 163 (48.1) | 82 (37.1) | |
| Men without hypertension | 125 (29.2) | 93 (27.4) | 68 (30.8) | |
| Age, years | 51.4 ± 8.0 | 53.2 ± 8.5 | 56.6 ± 8.8 | **<0.001** |
| Married | 323 (76.4) | 258 (76.6) | 166 (75.1) | 0.92 |
| ≥ Some college | 393 (92.5) | 311 (91.7) | 197 (88.7) | 0.26 |
| Full-time employment | 312 (72.7) | 238 (70.6) | 150 (68.5) | 0.52 |
| **Annual Household Income** | | | | |
| <$35,000 | 19 (4.8) | 21 (6.6) | 10 (4.7) | 0.08 |
| $35,000-100,000 | 115 (29.0) | 110 (34.6) | 83 (39.0) | |
| >$100,000 | 262 (66.2) | 187 (58.8) | 120 (56.3) | |
| *Clinical and biometric factors* | | | | |
| Anti-hypertensive medication use | 79 (18.4) | 75 (22.1) | 57 (25.7) | 0.09 |
| Cardiovascular medication use | 14 (3.3) | 18 (5.3) | 6 (2.7) | 0.21 |
| Diabetes diagnosis | 31 (7.2) | 26 (7.7) | 20 (9.0) | 0.72 |
| Cardiovascular disease | 18 (4.2) | 10 (3.0) | 12 (5.4) | 0.34 |
| *Self-reported health and mental health* | | | | |
| Depression | 74 (19.1) | 58 (17.4) | 23 (10.4) | **0.02** |
| Anxiety | 69 (17.8) | 44 (13.2) | 21 (9.5) | **0.01** |
| Self-reported excellent health | 107 (24.9) | 94 (27.7) | 61 (27.5) | 0.64 |
| *Blood pressure characteristics* | | | | |
| Baseline SBP | 119 ± 15 | 120 ± 15 | 121 ± 14 | 0.30 |
| Baseline DBP | 76 ± 9 | 76 ± 9 | 76 ± 8 | 0.89 |
| Daytime home SBP† | 123 ± 14 | 123 ± 13 | 121 ± 11 | 0.14 |
| Nighttime home SBP† | 122 ± 14 | 123 ± 13 | 121 ± 13 | 0.36 |
| SBP variability‡ | 0.11 | 0.10 | 0.09 | |
| DBP variability‡ | 0.12 | 0.10 | 0.10 | |
| Number of weeks with ≥ 1 home BP measurements | 5.1 (3.8) | 17.8 (7.3) | 41.8 (6.1) | **<0.001** |

Abbreviations: DBP – diastolic blood pressure; SBP – systolic blood pressure.

\* Data presented as N (%) or mean±SD.

† Daytime defined as 5am-5 pm, nighttime defined as 5 pm-5am.

‡ Coefficient of variability (standard deviation/mean).

an opportunity to further investigate how to promote HBPM adherence use among men with hypertension and in younger populations.

Our findings demonstrate a signal towards high adherence to HBPM among women with hypertension. This is a valuable insight given the sex disparities surrounding hypertension treatment, related comorbidities and cardiovascular outcomes. There is an increased risk of cardiovascular disease, myocardial infarction, and heart failure seen in women at lower systolic blood pressure ranges compared to men [38]. Women exhibit faster rates of blood pressure progression

**Table 3. Stratified multivariable logistic regression models evaluating hypertension and home blood pressure monitoring adherence.**

| | Women with Hypertension** | | Men with Hypertension** | |
| --- | --- | --- | --- | --- |
| | High adherence# | Gradual decrease# | High adherence# | Gradual decrease# |
| | aOR (95% CI) | aOR (95% CI) | aOR (95% CI) | aOR (95% CI) |
| Unadjusted Model | **2.24 (1.35, 3.72)** | 1.36 (0.86, 2.17) | 1.05 (0.62, 1.76) | 0.86 (0.52, 1.40) |
| Model 1* | 1.65 (0.97, 2.82) | 1.24 (0.77, 2.00) | 0.65 (0.37, 1.16) | 0.69 (0.41, 1.17) |
| Model 2† | 1.62 (0.93, 2.82) | 1.27 (0.77, 2.08) | 0.60 (0.33, 1.08) | 0.63 (0.37, 1.10) |
| Model 3‡ | **1.76 (1.01, 3.07)** | 1.19 (0.72, 1.96) | 0.76 (0.43, 1.37) | 0.68 (0.40, 1.16) |
| Model 4§ | **1.80 (1.04, 3.11)** | 1.21 (0.74, 1.98) | 0.70 (0.40, 1.25) | 0.70 (0.41, 1.20) |
| Model 5‖ | **1.85 (1.04, 3.28)** | 1.13 (0.68, 1.89) | 0.80 (0.44, 1.46) | 0.69 (0.39, 1.20) |

\* Model 1: adjusted for age.

† Model 2: adjusted for age, income.

‡ Model 3: adjusted for age, baseline systolic BP.

§ Model 4: adjusted for age, anxiety, depression.

‖ Model 5: adjusted for age, anxiety, depression, baseline systolic BP.

# Versus the "early discontinuation" group.

** Versus the same sex without hypertension.

with aging compared with men [22]. Additionally, women face additional barriers such as underrepresentation in clinical trials and underestimation of cardiovascular risk by clinical risk calculators, leading to higher rates of adverse drug effects and less aggressive/alternative treatments [10,39]. These disparities contribute to findings such as those of a recent meta-analysis which showed that women were less likely to be prescribed appropriate medical therapy for cardiovascular disease [40]. These increased risks have the potential to be offset through high adherence to HBPM, which could allow for close monitoring and titration of medications [40]. Future studies should examine the potential for the improvement of disparities in hypertension treatment and control if a highly adherent group such as women with hypertension are closely monitored with HBPM and given appropriate clinical guidance.

## Trajectory groups

Group-based trajectory modeling was used to derive three distinct trajectories of adherence to HBPM over the study period to examine adherence trajectories from a novel perspective. The high adherence trajectory group was the smallest and was characterized by those who achieved and maintained high levels of adherence throughout the one-year follow-up period. Given the overwhelming need for better blood pressure control nationally and internationally, our findings suggest that among certain patients, HBPM can be a tool that is accepted and used consistently.

## Maintaining adherence to HBPM

HBPM has been shown to be a convenient, acceptable, and effective method of diagnosing primary hypertension, white coat hypertension, and masked hypertension, and in helping to guide medication initiation and titration [41]. Particularly when combined with interventions such as remote telemonitoring, educational classes, and one-on-one counseling, HBPM is more effective than usual care at lowering blood pressure [18,42]. However, the major barrier that remains is how to best maximize adherence.

In our study, factors that were most strongly associated with high adherence were weakly associated with gradual decrease and inversely associated with early discontinuation. Throughout the one-year follow-up period in our study, over one-fifth of our participants maintained high adherence (>75%) to weekly HBPM. This high adherence group tended to be older in age, which

has been reported in prior studies [25,30,43]. This parallels findings that medication adherence may increase with older age, possibly due to older persons having greater severity of illness and fewer competing priorities such as young children and work, leading to awareness of chronic conditions and hence greater engagement with their health management [44].

Our BP adherence trajectory groups were found to have no significant differences in education level or full-time employment. This stands in contrast to the literature, including a survey study conducted in Texas, that found that persons with lower education level and income were significantly less likely to use blood pressure monitors [45]. Similarly, a Health-Styles survey found that regular HPBM was significantly associated with increased income, with a non-significant relationship seen with higher education level [26]. Lastly, a clustered randomized trial conducted in Minnesota found that higher education level, but not income level, predicted HBPM adherence [43].

Significantly more people in the early discontinuation group reported anxiety and depression, as compared to the gradual decrease and high adherence groups. It is possible that increased anxiety was a factor driving early discontinuation, perhaps due to worsening anxiety with blood pressure monitoring. In contrast, a randomized controlled trial conducted in England found that anxiety levels did not increase with HBPM [46]. In fact, a qualitative mixed methods study conducted with postpartum women found that self-management of blood pressure reduced anxiety [47]. However, a nationally representative survey of adults aged 50–80 years with hypertension or a blood pressure related health condition found that individuals who self-rated their mental health as "excellent or good" had a higher odds of regular HBPM, though anxiety/depression was not characterized [29].

Prior studies have identified other clinical factors associated with increased adherence to HBPM, including comorbidities of diabetes and coronary disease, being on antihypertensive medications, increased visceral obesity, having health insurance, and being more physically active [16,25,37]. These factors' effect on adherence was not directly examined in our cohort. Data are mixed regarding demographic associations with adherence. A cross-sectional study of NHANES suggested that individuals who are non-Hispanic Blacks and non-Hispanic Asians more frequently adhere to HBPM [16], whereas a HealthStyles survey found that non-Hispanic Blacks and Hispanics were less likely to be regular HBPM users compared to non-Hispanic whites [26]. Given that our participants were largely non-Hispanic White, our study is underpowered to derive any conclusion on racial or ethnic differences with HBPM adherence.

### Real-world implications

The growing knowledge of the value of HBPM combined with the known undertreatment of hypertension despite its devastating, long-term cardiovascular complications present an opportunity to intervene and help patients achieve better blood pressure control. A physician's recommendation of HBPM to patients has been found to increase rates of adherence not just to HBPM, but to medications as well [15]. The identification of subsets of patients less likely to adhere is important in identifying additional supports and resources.

### Use of digital heath tools

Our findings on adherence patterns are crucial in the context of the growing Internet of Medical Things, where ongoing patient engagement is important for effective healthcare. As real-time monitoring and internet-connected devices become more widely available, treatment plans can be tailored to meet the needs of individual patients to a degree never seen in history due to the sheer volume of personalized health data that can be collected and integrated.

At the same time, the use of artificial intelligence on these large data sets presents an opportunity to unlock new insights that traditional analysis may have otherwise missed [48].

### Strengths and limitations

This study has several strengths. A major strength was that we utilized robust data from the eFHS. Our study cohort was moderate in size, with women comprising a slight majority. Participants were not provided financial incentives, which potentially reduced participation bias. We utilized standardized, validated methods in describing adherence based on

existing literature, increasing the validity and reproducibility of our study findings. Moreover, our group-based trajectory modeling is a relatively novel and unique mode of estimating a change the adherence overtime. The use of such a model allows for the identification of subtle trends within the data, raising questions that can drive further research. For example, by detecting a "high adherence" trajectory group, we have identified a subset of participants who can potentially serve as a basis for further research on how best to implement HBPM.

Our study also has several limitations. First, the study cohort was predominantly White, English-speaking only, well-educated, generally healthy, and resided in New England. As such, the findings may not be generalizable to persons from different racial or ethnic backgrounds, regions, chronic conditions, and social determinants of health. Additionally, the study required participants to have an iPhone and utilize internet connection, omitting those who owned Androids and those who may not have the means to obtain a smartphone and stable internet. As wearable device data becomes increasingly used in clinical and epidemiological research, we acknowledge that study populations will be biased towards overrepresenting individuals with the means to access such technologies.

Moreover, we examined the association between sex and adherence over a relatively short 12-month period. This limits our study's generalizability of adherence in HBPM and long-term health benefits. Furthermore, these data are observational and cannot establish causal relations or rule out residual confounding. In the future, randomized studies would be needed to better elucidate factors driving increased adherence to HBPM. Finally, in our group-based trajectory modeling, we elected to choose a trajectory model that had a slightly suboptimal BIC score but was more clinically relevant. This has the potential to introduce statistical bias.

## Conclusion

In this middle-aged to older adult eFHS cohort, hypertension was associated with high adherence to HBPM in women with hypertension compared to those without. The targeted prescription of HBPM for women should be studied to see if it can reduce sex disparities in blood pressure control. Future studies should examine these associations in large populations of people with hypertension and consider further evaluation of how social determinants of health alter HBPM adherence.

## Supporting information

**S1 Table. Stratified multivariable logistic regression models evaluating hypertension and home blood pressure monitoring adherence in a 2-group trajectory model (high adherence vs low adherence) as a sensitivity analysis.**
(DOCX)

**S2 Table. Multivariable logistic regression models evaluating hypertension and home blood pressure monitoring adherence.**
(DOCX)

**S3 Table. Home blood pressure monitoring adherence by sex and hypertension status.**
(DOCX)

## Acknowledgments

All authors have read and approved the submission of the manuscript; the manuscript has not been published and is not being considered for publication elsewhere, in whole or in part, in any language, except as an abstract.

## Author contributions

**Conceptualization:** Tenes J. Paul, Apurv Soni, Honghuang Lin, Lara C. Kovell.

**Data curation:** Katherine Sadaniantz, Jean-Claude Asaker, Ziyue Wang.

**Formal analysis:** Tenes J. Paul, Ziyue Wang, Lara C. Kovell.

**Funding acquisition:** David D. McManus.

**Methodology:** Tenes J. Paul, Apurv Soni, Honghuang Lin, Lara C. Kovell.

**Supervision:** Lara C. Kovell.

**Writing – original draft:** Tenes J. Paul.

**Writing – review & editing:** Tenes J. Paul, Katherine Sadaniantz, Apurv Soni, Jean-Claude Asaker, Chathurangi H. Pathiravasan, Jordy Mehawej, Andreas Filippaios, Yuankai Zhang, Ziyue Wang, Chunyu Liu, Honghuang Lin, Joanne M. Murabito, David D. McManus, Lara C. Kovell.

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
