## [Decision Letter · Decision Letter 0]

9 Sep 2025

Dear Dr. Kovell,

Thank you for submitting your manuscript to PLOS ONE. After careful consideration, we feel that it has merit but does not fully meet PLOS ONE’s publication criteria as it currently stands. Therefore, we invite you to submit a revised version of the manuscript that addresses the points raised during the review process.

We look forward to receiving your revised manuscript.

Kind regards,

Buna Bhandari

Academic Editor

PLOS ONE

Journal Requirements:

The Framingham Heart Study (FHS) is supported by the National Heart, Lung, and Blood

Institute (NHLBI) of the National Institutes of Health and Boston University School of

Medicine, under NIH award 75N92019D00031. The electronic FHS research study was supported by

grant R01HL141434 and Robert Wood Johnson Award.

LK is supported by the National Heart, Lung, and Blood Institute through K23HL163450. DDM is supported by R01HL155343, R01HL141434, R33HL158541, U54HL143541 and U54HL143541-05S1, and UG3NS135168. AS is supported by U54HL143541 and U54HL143541-05S1, U01HL146382, and

UG3NS135168; AF is supported by American Heart Association AHA_18SFRN34110082; R01HL141434.

5. Please amend the manuscript submission data (via Edit Submission) to include author Joanne M. Murabito

6. Please amend your authorship list in your manuscript file to include author Murabito Murabito

7. We notice that your supplementary tables are included in the manuscript file. Please remove them and upload them with the file type 'Supporting Information'. Please ensure that each Supporting Information file has a legend listed in the manuscript after the references list.

Reviewers' comments:

Reviewer's Responses to Questions

**Comments to the Author**

1. Is the manuscript technically sound, and do the data support the conclusions?

Reviewer #1: Yes

Reviewer #2: Yes

Reviewer #3: Partly

2. Has the statistical analysis been performed appropriately and rigorously?

Reviewer #1: Yes

Reviewer #2: Yes

Reviewer #3: Yes

3. Have the authors made all data underlying the findings in their manuscript fully available?

Reviewer #1: Yes

Reviewer #2: Yes

Reviewer #3: Yes

4. Is the manuscript presented in an intelligible fashion and written in standard English?

Reviewer #1: Yes

Reviewer #2: Yes

Reviewer #3: Yes

Reviewer #1: 1. The introduction clearly explains that "women and older adults are more likely to adhere." It argues that the three-group model better reflects the typical patterns that real-life patients experience.

2. It would be helpful if authors add some details about the p-value for the difference between the groups for "Women with hypertension" and "Men with hypertension," which is currently reported as "0.04." If possible, provide separate p-values for these groups to clarify, or mention that this p-value shows the overall interaction.

3. The authors' study uses an Internet-connected blood pressure cuff, which is an example of the Internet of Medical Things (IoMT). Discussing the importance of IoMT adherence is a valuable addition. You may want to include a sentence like: "Our findings on adherence patterns are crucial in the context of the growing Internet of Medical Things (IoMT), where ongoing patient engagement is important for effective healthcare."

4. The authors justify their choice of the three-group model instead of the two-group model (which had a better Bayesian Information Criterion score), but this explanation could be expanded to show the clinical significance of the "gradual decrease" group.

5. Recommendations for Reference Additions: To strengthen the discussion, consider adding these references from the list provided, as they relate well to the technology in your study:

- To include in the Discussion or Limitations when discussing IoMT and data security:

- Rationale: The digital blood pressure device in your study is a clear example of an IoMT device. Citing sources on data protection in IoMT would show a forward-thinking approach and recognize the larger technological context of your research.

- Suggested Reference: Personal Data Protection Model in IoMT: Blockchain on Secured Bit-Count Transmutation Data Encryption Approach. Fusion: Practice & Applications, 16(1).

- To include in the Discussion when discussing future analysis of digital health data:

- Rationale: Your study uses advanced statistical methods. Mentioning how artificial intelligence could improve the analysis of such data would enhance the section on "Future studies."

- Suggested Reference: Artificial Intelligence in Improving Disease Diagnosis. A. Pati et al., “Artificial Intelligence in improving disease diagnosis,” Artificial Intelligence in Medicine and Healthcare, pp. 24–49, Jan. 2025. doi:10.1201/9781003508595-2 . The authors have created a high-quality manuscript that uses a sophisticated approach to explore real-world patterns of digital health engagement.

Reviewer #2: Thank you for the opportunity to review “Patterns of Adherence to Home Blood Pressure Monitoring Among Men and Women in the Electronic Framingham Heart Study". I enjoyed reading this insightful and informative submission and consider it a significant contribution to the literature on adults study. However please address some minor comments to enhance the quality of the manuscript.

Introduction

The introduction is good.

Methods

1) Indicate the study’s design with a commonly used term.

2) What was the total number of participants in the eFHS? Explain a paragraph about the study of eFHS.

3) What is meant by each of these groups? (Third Generation Cohort, the New Offspring Cohort, and the multiethnic Omni Group 2 Cohort.)

4) What is the meaning of the following phrase? (The eFHS data were accessed in July 2021. Authors did not have access to information that could identify individual participants during or after data collection. The eFHS cohort enrolled participants starting in June 2016 by inviting them during exam 3 (2016–2019) during regularly scheduled research center examinations.)

5) You state that the data was not available until 2021, please describe the relevant dates, including recruitment, exposure, follow-up, and data collection periods, relevant to your research period.

6) Explain how the study size was arrived at. What was the total number? How many people were eliminated?

7) Explain how missing data were addressed in Statistical methods.

Results

The results is good.

Discussion

The discussion is good - and draws the findings together in relation to previous research.

Given the proof of your hypothesis in this study and other previous studies, what practical and innovative suggestions do you have for future studies? (We hypothesize that women and older adults are more likely to adhere to HBPM )

Reviewer #3: It is natural that research using wearable or device-based data tends to have sample population biases, as access to technology and demographic characteristics often influence participation. At the same time, such data are becoming increasingly important for advancing epidemiological and clinical research. I appreciate that the authors acknowledge the potential limitations of their study population and their implications for generalizability. This work represents a valuable attempt to investigate adherence patterns using innovative modeling approaches. At the same time, it is important for the authors to help readers remain aware of how these sample-related biases may influence the interpretation of findings and the conclusions drawn.

**Do you want your identity to be public for this peer review?** For information about this choice, including consent withdrawal, please see our Privacy Policy

Reviewer #1: No

Reviewer #2: No

Reviewer #3: **Yes: ** Jianlei Gu

---

## [Author Response · Author response to Decision Letter 1]

15 Oct 2025

The following responses are also included in the "Response to Reviewers" file. The reviewers' comments are listed, followed by our response. The formatting is easier to follow on that separate document.

Comments from the Academic Editor

We have gone through the manuscript and adjusted the formatting to match the style templates listed above.

The “Funding” section of the manuscript has now been updated to match the “Financial Disclosure” section in the submission portal.

3. Thank you for stating the following financial disclosure: The Framingham Heart Study (FHS) is supported by the National Heart, Lung, and Blood Institute (NHLBI) of the National Institutes of Health and Boston University School of Medicine, under NIH award 75N92019D00031. The electronic FHS research study was supported by grant R01HL141434 and Robert Wood Johnson Award. LK is supported by the National Heart, Lung, and Blood Institute through K23HL163450. DDM is supported by R01HL155343, R01HL141434, R33HL158541, U54HL143541 and U54HL143541-05S1, and UG3NS135168. AS is supported by U54HL143541 and U54HL143541-05S1, U01HL146382, and UG3NS135168; AF is supported by American Heart Association AHA_18SFRN34110082; R01HL141434. Please state what role the funders took in the study. If the funders had no role, please state: "The funders had no role in study design, data collection and analysis, decision to publish, or preparation of the manuscript." If this statement is not correct you must amend it as needed. Please include this amended Role of Funder statement in your cover letter; we will change the online submission form on your behalf.

Thank you for pointing this out. The funders had no role in study design, so we have added the quoted statement above to the “Funding” section of the manuscript and copied it to the cover letter as well.

The ethics statement was moved from the “Acknowledgement” section to the “Methods” section. It does not appear in any other section.

5. Please amend the manuscript submission data (via Edit Submission) to include author Joanne M. Murabito

This change was made

6. Please amend your authorship list in your manuscript file to include author Murabito Murabito

We added Joanne M. Murabito as described above. Thank you for pointing out this oversight.

7. We notice that your supplementary tables are included in the manuscript file. Please remove them and upload them with the file type 'Supporting Information'. Please ensure that each Supporting Information file has a legend listed in the manuscript after the references list.

The supplementary tables have been removed from the manuscript. Only the tables’ titles and legends remain.

Thank you.

Comments from Reviewer #1

1. It would be helpful if authors add some details about the p-value for the difference between the groups for "Women with hypertension" and "Men with hypertension," which is currently reported as "0.04." If possible, provide separate p-values for these groups to clarify, or mention that this p-value shows the overall interaction.

Thank you for pointing this out. For additional clarity, the following tables were combined into a supplemental table.

Early discontinuation Gradual decrease High adherence p-value

Women with hypertension 42 (42.0) 46 (55.4) 38 (53.5) 0.14

Men with hypertension 58 (58.0) 37 (44.6) 33 (46.5)

Early discontinuation Gradual decrease High adherence p-value

Women with hypertension 42 (17.1) 46 (22.0) 38 (31.7) 0.007

Women without hypertension 203 (82.9) 163 (78.0) 82 (68.3)

Early discontinuation Gradual decrease High adherence P-value

Men with hypertension 58 (31.7) 37 (28.5) 33 (32.7) 0.75

Men without hypertension 125 (68.3) 93 (71.5) 68 (67.3)

2. The authors' study uses an Internet-connected blood pressure cuff, which is an example of the Internet of Medical Things (IoMT). Discussing the importance of IoMT adherence is a valuable addition. You may want to include a sentence like: "Our findings on adherence patterns are crucial in the context of the growing Internet of Medical Things (IoMT), where ongoing patient engagement is important for effective healthcare."

This is a fascinating and important insight. We have added a paragraph to the end of the “Real world implications” subsection of the “Discussion” describing the importance of our findings in the context of the IoMT.

3. The authors justify their choice of the three-group model instead of the two-group model (which had a better Bayesian Information Criterion score), but this explanation could be expanded to show the clinical significance of the "gradual decrease" group.

Additional statements were added to the “Group based trajectory modeling” subsection of the “Methods” section about this. The 3-group model is more inclusive of the range of adherence encountered in clinical practice: some patients will follow instructions long term, other will never adhere, and others will initially demonstrate adherence, but may ultimately cease to adhere.

4. Recommendations for Reference Additions: To strengthen the discussion, consider adding these references from the list provided, as they relate well to the technology in your study:

4.1. To include in the Discussion or Limitations when discussing IoMT and data security:

4.1.1. Rationale: The digital blood pressure device in your study is a clear example of an IoMT device. Citing sources on data protection in IoMT would show a forward-thinking approach and recognize the larger technological context of your research.

4.1.2. Suggested Reference: Personal Data Protection Model in IoMT: Blockchain on Secured Bit-Count Transmutation Data Encryption Approach. Fusion: Practice & Applications, 16(1).

4.2. To include in the Discussion when discussing future analysis of digital health data:

4.2.1. Rationale: Your study uses advanced statistical methods. Mentioning how artificial intelligence could improve the analysis of such data would enhance the section on "Future studies."

4.2.2. Suggested Reference: Artificial Intelligence in Improving Disease Diagnosis. A. Pati et al., “Artificial Intelligence in improving disease diagnosis,” Artificial Intelligence in Medicine and Healthcare, pp. 24–49, Jan. 2025. doi:10.1201/9781003508595-2 .

Thank you for both of these discussion points. We added a section to the end of the “discussion” section entitled, “Use of digital health tools.” In this section, we discuss our findings in the context of the Internet of Medical Things and how AI may impact both the handling of personal health data and studies on digital health tool usage in the future.

Comments from Reviewer #2

1. Indicate the study’s design with a commonly used term.

Thank you for your thoughtful comments and questions. Please see the point-by-point responses below.

This is a prospective cohort study. This was added to the first paragraph of the “Methods” section

2. What was the total number of participants in the eFHS? Explain a paragraph about the study of eFHS.

Total number of participants in eFHS were 1918.

We have added a paragraph briefly providing an overview of eFHS in the methods section.

3. What is meant by each of these groups? (Third Generation Cohort, the New Offspring Cohort, and the multiethnic Omni Group 2 Cohort.)

These cohorts are the third generation of the original Framingham Heart Study cohort, spouses of the offspring of the original Framingham Heart Study cohort, and members of the Framingham community from underrepresented ethnic groups, respectively.

4. What is the meaning of the following phrase? (The eFHS data were accessed in July 2021. Authors did not have access to information that could identify individual participants during or after data collection. The eFHS cohort enrolled participants starting in June 2016 by inviting them during exam 3 (2016–2019) during regularly scheduled research center examinations.)

Clarification was added in the methods section.

5. You state that the data was not available until 2021, please describe the relevant dates, including recruitment, exposure, follow-up, and data collection periods, relevant to your research period.

Participants were recruited by the Framingham Heart Study team between 2016 and 2019. All follow up and data collection happened during this period. In 2021, we (the authors of the current manuscript) obtained access to the data analyzed in this manuscript from the Framingham Heart Study team.

6. Explain how the study size was arrived at. What was the total number? How many people were eliminated?

Among the 1918 eFHS enrollees, 1125 participants chose to use a digital blood pressure device. Those who did not record at least 1 blood pressure reading were excluded, leaving 990 participants.

7. Explain how missing data were addressed in Statistical methods.

We use Chi‐square test, ANOVA test, and logistic regression in statistical analysis. Those test/regression handle missing data by discarding any observation that had a missing value for any variable. A sentenced describing this was added to the statistical analysis section of the “Methods.”

8. Given the proof of your hypothesis in this study and other previous studies, what practical and innovative suggestions do you have for future studies? (We hypothesize that women and older adults are more likely to adhere to HBPM )

We added a section to the “discussion” entitled, “Use of digital health tools,” where we discuss the implications of the wide use of commercially available digital health tools and the potential insights that can be gained from the large amount of data produced with these devices. We highlight that the use of artificial intelligence in future studies using these large datasets may have the potential to unlock new insights.

Comments from Reviewer #3

1. This work represents a valuable attempt to investigate adherence patterns using innovative modeling approaches. At the same time, it is important for the authors to help readers remain aware of how these sample-related biases may influence the interpretation of findings and the conclusions drawn.

We agree that this is an important point. We added a sentence to the end of paragraph 2 of the “Strengths and Limitations” section acknowledging this.

---

## [Decision Letter · Decision Letter 1]

7 Nov 2025

Patterns of Adherence to Home Blood Pressure Monitoring Among Men and Women in the Electronic Framingham Heart Study

PONE-D-25-37475R1

Dear Dr. Kovell,

We’re pleased to inform you that your manuscript has been judged scientifically suitable for publication and will be formally accepted for publication once it meets all outstanding technical requirements.

Kind regards,

Dr Buna Bhandari

Academic Editor

PLOS ONE

Additional Editor Comments (optional):

Reviewers' comments:

Reviewer's Responses to Questions

**Comments to the Author**

Reviewer #1: All comments have been addressed

Reviewer #2: All comments have been addressed

2. Is the manuscript technically sound, and do the data support the conclusions?

Reviewer #1: Yes

Reviewer #2: Yes

3. Has the statistical analysis been performed appropriately and rigorously?

Reviewer #1: Yes

Reviewer #2: Yes

4. Have the authors made all data underlying the findings in their manuscript fully available?

Reviewer #1: Yes

Reviewer #2: Yes

5. Is the manuscript presented in an intelligible fashion and written in standard English?

Reviewer #1: Yes

Reviewer #2: Yes

Reviewer #1: The authors have addressed all comments and suggestions, which has strengthened the manuscript.

1. The authors expanded the explanation for choosing a three-group model instead of a two-group model in the Methods section. They highlighted the clinical importance and inclusiveness of the "gradual decrease" group.

2. A new section called "Use of Digital Health Tools" was added to the Discussion. This section links the findings on adherence to the larger context of the Internet of Medical Things (IoMT) and suggests that Artificial Intelligence (AI) could be used to analyze large digital health datasets for new insights.

3. The study design is clearly stated as a prospective cohort study. The authors clarified the total number of participants in the eFHS (N=1918), the exclusion criteria leading to N=990 for analysis, the specific cohort groups (Third Generation, New Offspring, Omni Group 2), and the relevant dates for data collection and access.

4. The authors updated the Statistical Methods section to clearly explain that they addressed missing data by removing any observation with a missing value for any variable used in the Chi-square test, ANOVA test, or logistic regression.

5. The authors updated Supplementary Table 3 to include separate p-values that show the differences in adherence patterns, especially for women with and without hypertension (p=0.007) and men with and without hypertension (p=0.75), improving transparency.

6. The authors added a sentence to the "Strengths and Limitations" section to acknowledge that sample-related biases, such as those from needing specific technology, can influence how the findings are interpreted and applied.

Reviewer #2: (No Response)

**Do you want your identity to be public for this peer review?** For information about this choice, including consent withdrawal, please see our Privacy Policy

Reviewer #1: No

Reviewer #2: No

---

## [Editor Report · Acceptance letter]

PONE-D-25-37475R1

PLOS One

Dear Dr. Kovell,

I'm pleased to inform you that your manuscript has been deemed suitable for publication in PLOS One. Congratulations! Your manuscript is now being handed over to our production team.

Kind regards,

on behalf of

Dr. Buna Bhandari

Academic Editor

PLOS One